# INFORMATION SUBTRACTION: LEARNING REPRESENTATIONS FOR CONDITIONAL ENTROPY

## ABSTRACT

The representations of conditional entropy and conditional mutual information are significant in explaining the unique effects among variables. The previous works based on conditional contrastive sampling have successfully eliminated information about discrete sensitive variables, but have not yet addressed continuous cases. This paper introduces a framework of Information Subtraction capable of representing arbitrary information components between continuous variables. We implement a generative-based architecture that outputs such representations by simultaneously maximizing an information term and minimizing another. The results highlight the representations' ability to provide semantic features of conditional entropy. By subtracting sensitive and domain-specific information, our framework effectively enhances fair learning and domain generalization.

## 1 INTRODUCTION

In complex systems, the variables could be entangled and correlated, thereby providing mutual information to each other. Such uncertainties and information can be quantified using information-theoretic measures depicted in Figure 1. These measurements help identify the relationships between variables, such as correlation and Granger causality (Pearl, 2009). Beyond merely recognizing such relationships, many works aim to further explain and represent them (Yao et al., 2021; Xu et al., 2023), which enhance our understanding and control of the system. These representation learning approaches generate representations that maximize their information about the targets, as they must be capable of accurately recovering the targets during the reconstruction part (Kingma & Welling, 2013; Clark et al., 2019). However, while most methods effectively represent entropy (the entire circle in Figure 1(a)), few address the representation of other information terms like conditional mutual information and conditional entropy.

The representation of conditional mutual information is significant because it can reveal the unique effect of certain factors on the target that other factors cannot provide. For example, representing the unique effect of funding on scholars' publications can guide policy suggestions, such as terminating funding that shows insignificant boosting. Additionally, removing the effects of sensitive factors helps to create fair and unbiased representations. For instance, we may want to eliminate biases in employee performance evaluations due to discriminatory factors such as gender and race. We can achieve this objective through the representation of conditional entropy.

To represent conditional entropy $H(Y|X) = H(Y) - I(X;Y)$, we should overcome the challenge to selectively include information about $Y$ while excluding the information about $X$. This requires the architecture to optimize the representation by simultaneously maximizing and minimizing certain information terms during training. (Ma et al., 2021) bypasses this challenge through conditional sampling that selects samples with same conditions to be the input of neural networks. However, this sampling method is limited to cases with continuous conditional variables, where (Tsai et al.) proposes to apply clustering algorithms to select samples with similar conditions.

To address the challenge of selectively maximizing and minimizing different information terms, we introduce the concept of Information Subtraction. It aims to subtract specific information from a representation, which we implement through a generative-based architecture. The resulting representation retains the desired information within the target scope $Y$ and filters out the undesired information $X$ beyond it. Furthermore, by iteratively applying Information Subtraction, we can rep-

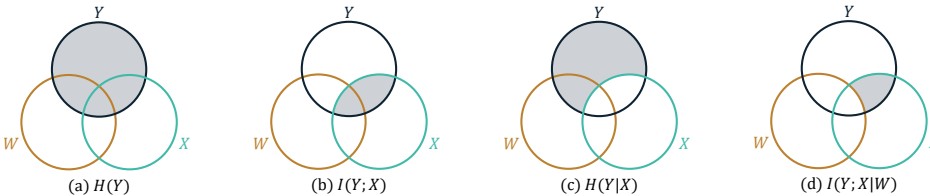

(a) $H(Y)$     (b) $I(Y;X)$     (c) $H(Y|X)$     (d) $I(Y;X|W)$

Figure 1: Venn Diagram of entropy $H(Y)$, mutual information $I(Y;X)$, conditional entropy $H(Y|X)$, and conditional mutual information $I(Y;W|X)$.

resent any segment within the Venn Diagram (Figure 1(b), (c), (d)) independently, not limited to just conditional entropy.

Our framework consists of a generator neural network with two discriminators which provides stable estimation of the information terms. The generator's objective is to maximize the information estimation from the first discriminator while minimizing that from the second. We demonstrate that this representation for conditional entropy can decompose signals and encode semantic meanings. By subtracting sensitive and domain-specific information, our framework effectively enhances fair learning and domain generalization.

We state the contributions of our work as follows:

- We highlight the significance and applications of conditional representation learning.
- We introduce the Information Subtraction framework and demonstrate its capacity to decompose signals and encode semantic meanings.
- We propose a generative-based framework for Information Subtraction to generate conditional representations.
- We demonstrate the effectiveness of the representation of conditional entropy in fair learning and domain generalization.

## 2 PROBLEM DESCRIPTIONS

Define $Y$ the target variable, and $X$ the conditional variable. We assume that $X$ can only provide part of the information about $Y$, in which $H(Y) > I(Y;X)$ (red circle in Figure 2) and $H(Y|X) > 0$. Our goal is to generate a set of implicit representations $Z = \{Z^1, Z^2, ..., Z^m\}$ to maximize $I(Y;X,Z)$ (black circle), and expand it to approximate the entropy $H(Y)$ (the entire yellow box). Besides, we also aim at minimizing the information of $X$ in $Z$, hence the generated $Z$ should represent the information in the shaded region.

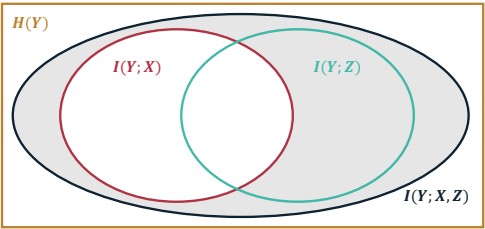

Figure 2: The Venn Diagram illustrates the information provided to $Y$ by $X$ (red), $Z$ (blue), and $\{X, Z\}$ (black). Our objective is to generate $Z$ that represents the shaded region.

If we can identify a variable $Z$ that perfectly provides this information, we can infer that $Z$ is a representation of $H(Y|X) = H(Y) - I(X;Y)$. More specifically, $Z$ captures the information relating to $Y$ that is not provided by $X$. Considering

$$\sup_Z I(Y;Z|X) = H(Y|X), \tag{1}$$

our objective is to learn the distribution of representation $Z$ given target $Y$, $P_{Z|Y}$, that maximizes the conditional mutual information $I(Y; Z|X)$ and makes it close to $H(Y|X)$:

$$\max_{P_{Z|Y}} I(Y; Z|X). \tag{2}$$

Notice that

$$I(Y; Z|X) = I(Y; X, Z) - I(Y; X), \tag{3}$$

the second term is considered constant when optimizing $Z$, and hence the objective becomes:

$$\max_{P_{Z|Y}} I(Y; X, Z) \tag{4}$$

Recall that $H(Y|X)$ is the information of $Y$ that $X$ cannot provide, and thus $Z$ should not include any information about $X$. Therefore, this objective is not suitable here as it does not restrict $Z$ to eliminate the information of $X$. Hence, our objective with a restriction term should be:

$$\max_{P_{Z|Y}} I(Y; X, Z)$$
$$\text{subject to } I(X; Z) = 0 \tag{5}$$

In this work, the training objective is released with a trade-off hyperparameter $\lambda$, to make $Z$ represent the conditional entropy $H(Y|X)$ without carrying information from $X$:

$$\max_{P_{Z|Y}} I(Y; X, Z) - \lambda I(X; Z) \tag{6}$$

## 3 RELATED WORKS

Many frameworks aim to maximize the information content of their representations for target variables, assuming that target observations are driven or caused by a set of unobserved latent signals or representations. These frameworks are types of self-supervised learning (Zhang et al., 2023), which inputs target observations itself to generate representations without auxiliary supervising labels or information. Within this domain, two common approaches are generative-based and contrastive-based methods.

The objective of generative-based approaches (Hyvärinen et al., 2008; Hyvarinen & Morioka, 2017; Hjelm et al., 2018; Klindt et al., 2020; Wu et al., 2020) is to optimize the posterior distributions of representations given the observations (encoding) and vice versa (decoding). The optimal encoder, which maximizes the mutual information between representation and observations, is attained through various methods (Wu et al., 2020; Hjelm et al., 2018), including Variational Autoencoder (VAE) (Kingma & Welling, 2013; Louizos et al., 2017; Yao et al., 2021) and Dynamical Component Analysis (DCA) (Clark et al., 2019; Bai et al., 2020; Meng et al., 2022). Although these methods ensure the identifiability of the latent variables, their structures do not specifically aim to represent conditional information, thus motivating our work to achieve this goal.

Following the idea that similar samples should have similar representations, contrastive-based approaches (Ma et al., 2021) focus on contrasting the representations of positive or similar pairs of observations against negative or dissimilar pairs. They aim to generate a contrastive representation $Z$ of the target $Y$ that maximizes the conditional mutual information $I(Z; Y|X)$. This is achieved through conditional sampling: $X$ is first sampled, followed by the sampling of positive and negative pairs $Y$ that share the same $X$. This approach performs outstandingly with discrete and categorical $X$, but not with continuous cases. To address this limitation, (Tsai et al.) suggests identifying samples with similar $X$ values using clustering algorithms. Rather than performing conditioning at the sampling stage, our method manipulates the information in the training objective and effectively handles continuous $X$ at the training stage.

Our work performs information maximization and minimization simultaneously, sharing a similar objective to that of (Chen et al., 2016; Moyer et al., 2018; Kim et al., 2019; Zhu et al., 2021; Ragonesi et al., 2021). These works are capable of eliminating the information of biasing or sensitive terms from the representations, and hence can be applied to fair learning and domain generalization problems. Besides, there are other works that perform operations similar to information minimization,

such as the use of Maximum Mean Discrepancy (MMD) penalty (Louizos et al., 2015), $\beta$-VAE (Higgins et al., 2017), and maximum cross-entropy (Alvi et al., 2018). While we share similar ~~architectures~~ objectives with these works, their structures are not designed for conditional representations. We build upon and expand their frameworks to perform representation learning on conditional entropy and conditional mutual information.

# 4 ARCHITECTURE

Based on our previous work (**?**), We have proposed an expanded architecture to satisfy the objective in Equation 6 that simultaneously maximize $I(Y; X, Z)$ and minimize $I(X; Z)$. As shown in Figure 3, our architecture consists of three neural networks: one generator that outputs $Z$ and two discriminators to estimate $I(Z, X; Y)$ and $I(Z; X)$ respectively.

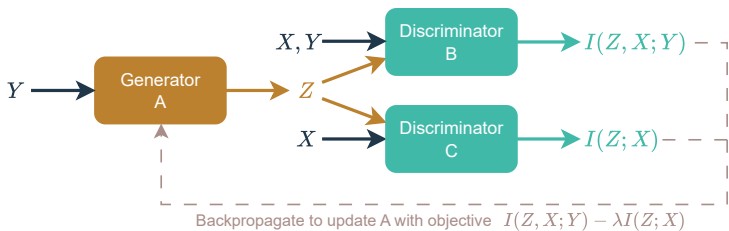

Figure 3: The illustration of the architecture. $Y$ is the target variable, $X$ is the conditional variable. Generator A inputs $Y$ and outputs its representation $Z$, which is then fed into Discriminator B and C to estimate the loss function to be back-propagated.

## 4.1 GENERATOR

We assume that $Z$ is implicitly involved in the system and provides information of $Y$ that $X$ cannot provide. Suppose the information of $Y$ is provided by $\{X, Z\}$ (if $Z$ can ideally represent $H(Y|X)$), reversely, the information of $Z$ should be covered by (a part of) $Y$. Therefore, $Z$ could be described as a function of $Y$, which is implemented by the generator. We do not specify the architecture for the generator. It could be a normal feedforward neural network, RNN, or CNN, as long as the generators can output adequate representations $Z$ for the input data types.

## 4.2 DISCRIMINATOR

The discriminator provides a measurement of $I(Y; X, Z)$ and $I(Y; Z)$, where $Z$ is the output of the generator. Consider $Y$ and $\{X, Z\}$ as two sets of (possibly high-dimensional) random variables, and their mutual information is equivalent to the Kullback-Leibler (KL-) divergence between their marginal and joint distributions,

$$
\begin{aligned}
I(Y; X, Z) &= D_{KL}\left(\mathbb{P}_{\boldsymbol{YXZ}} \,\|\, \mathbb{P}_{\boldsymbol{Y}} \bigotimes \mathbb{P}_{\boldsymbol{XZ}}\right) \\
&= \sup_{T:\boldsymbol{Y}\times\boldsymbol{X}\times\boldsymbol{Z}\to\mathbb{R}} \mathbb{E}_{\mathbb{P}_{\boldsymbol{YXZ}}}[T] - \log\left(\mathbb{E}_{\mathbb{P}_{\boldsymbol{Y}}\otimes\mathbb{P}_{\boldsymbol{XZ}}}[e^T]\right)
\end{aligned} \tag{7}
$$

where $D_{KL}$ denotes the KL-divergence of two distributions, and $\mathbb{P}_{\boldsymbol{Y}}$ denotes the distribution of $Y$. The lower bound of the KL-divergence is provided by the Donsker-Varadhan representation in Equation 7, where $T(Y, X, Z)$ is any class of function. We follow MINE (Belghazi et al., 2018) to implement function $T$ by a neural network with parameters $\Theta$ and maximize the lower bound of mutual information $I_\Theta$ by searching for the optimized function $T_\theta$:

$$
I(Y; X, Z) \geq I_\Theta(Y; X, Z) \tag{8}
$$

$$
I_\theta(W, Y) = \max_{\theta\in\Theta} \mathbb{E}_{\mathbb{P}_{WY}}[T_\theta] - \log\left(\mathbb{E}_{\mathbb{P}_W\otimes\mathbb{P}_Y}\left[e^{T_\theta}\right]\right) \tag{9}
$$

The optimization is done by stochastic gradient descent (SGD). The available gradient is essential in our architecture because one of the inputs of MINE, $Z$, is exactly the output of the generator. Hence,

the generator and discriminator are fused together by $Z$, which acts as the bridge to back-propagate the gradient from the discriminator to the generator and thus update the parameters in both neural networks. A benefit of this generator-discriminator structure is that it does not make any requirement or assumption about the distribution of $X$, $Z$, or $Y$.

In this work, we use an advanced variant of MINE, named SMILE (Song & Ermon, 2020), which achieves more convergent and accurate results by implementing clipping in gradient descent to stabilize the training procedures. Obtaining stable estimations for both $I(Y; X, Z)$ and $I(Y; Z)$ is important, as the generator updates its parameters according to the gradient direction, aiming to maximize the former while minimizing the latter. It is worth mentioning that some other possible models for the discriminators include (Mukherjee et al., 2020; Cheng et al., 2020). We choose SMILE in this work simply because of its best performance.

### 4.3 MAXIMIZING CONDITIONAL MUTUAL INFORMATION

---
**Algorithm 1** Information Subtraction
---
**Input:** conditional variable $X$, target $Y$, $n_1 < n_2, n_3$;
**Output:** representation $Z$;
 1: Initialize neural networks: generator $N_A$, reconstructor $N_B$, discriminator $N_C$ for $I(Y; X, Z)$, discriminator $N_D$ for $I(X; Z)$;
 2: **for** $i = 1, 2, \cdots, n_2$ **do**
 3:     Sample batch $x$, $y$;
 4:     $z \leftarrow N_A(y)$;
 5:     **if** $i < n_1$ **then**
 6:         Update $N_A$ w.r.t. minimizing $l_1 = \|N_B(z) - y\|$;
 7:     **else**
 8:         Update $N_A$ w.r.t. minimizing $l_2 \leftarrow N_D(x, z) - N_C(x, y, z) = I(X; Z) - I(Y; X, Z)$;
 9:     **end if**
10:     **for** $j = 1, 2, \cdots, n_3$ **do**
11:         Sample batch $x$, $y$;
12:         $z \leftarrow N_A(y)$;
13:         Update $N_C$ w.r.t. minimizing $l_3 \leftarrow -N_C(x, y, z) = -I(Y; X, Z)$;
14:         Update $N_D$ w.r.t. minimizing $l_4 \leftarrow -N_D(x, z) = -I(X; Z)$;
15:     **end for**
16: **end for**

---

Algorithm 1 shows how our architecture in Figure 3 works. We split the training into two stages. In the pre-training stage (first $n_1$ epochs), in order to ensure that $Z$ carries the information of $Y$, the generator's parameters are updated by minimizing the reconstruction (performed by $N_B$) loss to $Y$. When the result converges, we start to minimize $I(X; Z)$ (loss = $I(X; Z)$) while maximizing $I(Y; X, Z)$ (loss = $-I(Y; X, Z)$). Iteratively updating generator's parameters towards the direction of minimizing $I(X; Z)$ and $-I(Y; X, Z)$ can gradually achieve our objective. To provide stable and accurate information estimates, we update the discriminators' parameters to learn $I(Y; X, Z)$ and $I(X; Z)$ in every epoch.

The generator and discriminator are fused together, forming a whole neural network that trains both components simultaneously, where $Z$ could be viewed as an intermediate layer. While training together by gradient descent, the generator is trained to improve the generation of $Z$ to maximize the objective, while the discriminators are trained to provide more accurate estimations of $I_\theta$ given the $Z$ from the generator. Therefore, the structure pursues the objective:

$$\max_{\theta_1 \in \Theta_1, \, \theta_2 \in \Theta_2, \, P_{Z|Y}} I_{\theta_1}(Y; X, Z) - \lambda I_{\theta_2}(X; Z) \tag{10}$$

## 5 APPLICATION SCENARIOS AND RESULTS

In this section, we have constructed two synthetic examples to illustrate how our architecture generates representations for conditional entropy and conditional mutual information, and how it captures the relationships among variables. Additionally, we examine the framework's efficacy on real-world datasets in fair learning and domain generalization.

## 5.1 REPRESENTING RELATIONSHIPS

Figure 4(a) and (b) show a synthetic predator-prey model, with one predator species, wolves, and two prey species, sheep and rabbits. The prey species share grass as the common food source. In total, there are four distinct species in the ecosystem, and their populations evolve according to the Lotka–Volterra model (Lotka, 1925):

$$\frac{dW(t)}{dt} = W(t)(-a_0 + a_1 S(t) + a_2 R(t))$$
$$\frac{dS(t)}{dt} = S(t)(b_0 - b_1 W(t) + b_2 G(t))$$
$$\frac{dR(t)}{dt} = R(t)(c_0 - c_1 W(t) + c_2 G(t)) \tag{11}$$
$$\frac{dG(t)}{dt} = G(t)(d_0 - d_1 S(t) - d_2 R(t)),$$

where $W(t)$, $S(t)$, $R(t)$, and $G(t)$ denote the population of wolves, sheep, rabbits, and grass at time $t$, respectively. The parameter settings and implementation details are provided in Appendix A. From the equations, we observe that $S$ and $R$ cover the information of $G$. Table 1 demonstrates that the generated representation $Z$ contains information about $H(G|S)$ in this example.

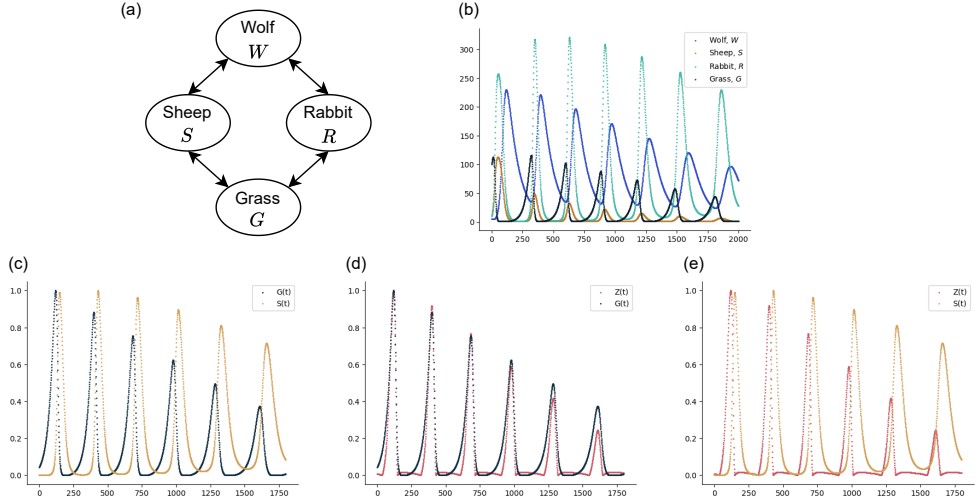

Figure 4: (a) The relationship diagram between four species. (b) The dynamics of Lotka–Volterra model. (c) The dynamics of $S$ and $G$ over $t$. (c) The dynamics of $Z$ and $G$ over $t$. (c) The dynamics of $Z$ and $S$ over $t$.

Table 1: Experiment results: Information quantities among $S$, $G$, and the generated $Z$.

| $I(S;G)$ | $H(G)$ | $H(G|S)$ | $I(Z;G)$ | $I(Z;S)$ | $I(Z,S;G)$ | $I(Z;G|S)$ |
|---|---|---|---|---|---|---|
| 0.89 | 3.28 | 2.39 | 2.06 | 0.28 | 2.33 | 1.44 |

We observe that $S$ contributes only 0.89 bits of information to $G$, whereas the amount of total entropy $H(G)$ is 3.28 bits. The generated representation $Z$ provides an additional 1.44 bits of information to $G$ that $S$ cannot provide, while containing only 0.28 bits of information about $S$. This result shows that our architecture is capable of performing Information Subtraction by generating $Z$ that covers and represents most of the information in $H(G|S)$, and contain minimal information about $G$.

So how can the representation capture a relationship? Here, we define the relationship from $S$ to $G$ as the semantic features that both variables share ($I(S;G)$). Semantic feature means what can we

infer about the corresponding $G$ given the value of a sample of $S$. Define the features of $G$ provided uniquely by $S$, $R$, and jointly provided by $\{S, R\}$ as $f_{S \to G}$, $f_{R \to G}$, and $f_{S,R \to G}$ respectively. Then,

$$G = g(f_{S \to G}, f_{R \to G}, f_{S,R \to G}), \tag{12}$$

where $g$ is a simple function that aggregates these features to obtain $G$. Hence, we can define the relationship from $S$ to $G$ as the union of $\{f_{S \to G}, f_{S,R \to G}\}$. In Figure 4(c) we observe that $S$ provides certain features to $G$, including:

- When $S$ is at its maximum, we can infer that $G$ is at its minimum.
- When $S$ is within its middle range, we can infer that $G$ is within its lower or middle range.
- When $S$ is within its lower range, we can infer that $G$ is within its middle range.
- When $S$ is at its minimum, we can infer that $G$ is at its maximum.

Although $S$ provides many informative features, some other information is still missing. Specifically, it can only indicate the trends but not the exact value of $G$. In Figure 4(d), we observe that the generated representation $Z$ provides additional features to $G$ that $S$ does not, including:

- When $Z$ is at its maximum, we can infer that $G$ is at its maximum, along with the corresponding peak value.
- When $Z$ is within its upper range, we can infer that $G$ is within its upper range, along with the corresponding value.

Representing the semantic meanings of conditional entropy is significant when we aim to eliminate the information about certain variables from the target. This will be further demonstrated in the following cases in Section 5.3, 5.4, and 5.5.

## 5.2 INFORMATION SUBTRACTION

In the previous case, we only focus on the conditional entropy between two variables. For multivariate scenarios, we can apply Information Subtraction to generate representations for any sector in the Venn Diagram in Figure 5.

In the case of three variables, we can first start with the outermost sectors and generate $Z_1$, $Z_2$, and $Z_3$ for the conditional entropies $H(X|Y, W)$, $H(Y|X, W)$, and $H(W|X, Y)$, respectively. Subsequently, we can again perform Information Subtraction on $Z_1$, $Z_2$, $Z_3$ to generate $Z_4$, $Z_5$, and $Z_6$ for conditional mutual information terms. Finally, by conducting Information Subtraction on the previous $Z$, we can obtain the representation $Z_7$ of trivariate mutual information, $I(X; Y; W)$. A detailed demonstration of this process is provided in the Appendix B.

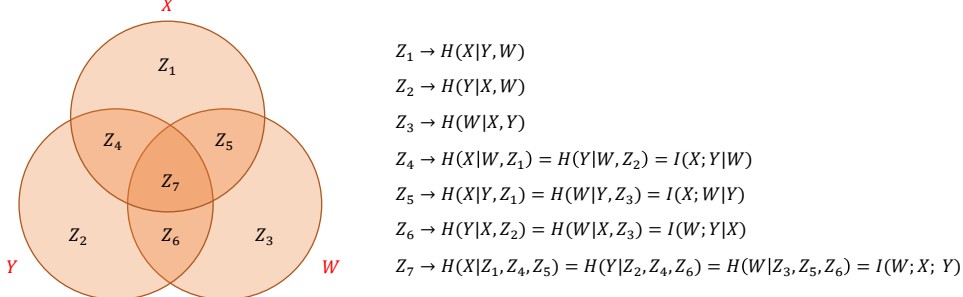

Figure 5: Information Subtraction.

The benefit of Information Subtraction is that the representation can effectively include the information within the sector while excluding any information outside. The idea here is to decompose the entangled signals $\{X, Y, W\}$ into mutually exclusive sectors represented by $\{Z_1, ..., Z_7\}$. With the property of additivity, we can aggregate representations from arbitrary sectors together to represent different information terms. For example, the combination $\{Z_1, Z_5\}$ represents $H(X|Y)$, and $\{Z_4, Z_7\}$ represents $I(X; Y)$.

### 5.3 SYNTHETIC CASE FOR FAIR LEARNING

To illustrate the application of conditional entropy representations in fair learning, we begin with a synthetic example, followed by an analysis using a real dataset. Suppose the academic performances of researchers from three countries follow the pattern in Figure 6(a):

$$Y = VW,$$
$$p(w|x=0) = N(0.7, 0.05), \quad p(w|x=1) = N(0.5, 0.1), \quad p(w|x=2) = N(0.3, 0.07), \quad (13)$$

where $Y$ is academic performance of the researcher, $X$ is their countries, $V$ is intelligence, and $W$ is research training experience. Suppose $Y$ is determined solely by $V$ and $W$, and is independent of $X$. Furthermore, a researcher's intelligence $V$ is not determined by which country $X$ they come from, but the research training experience $W$ is.

In this example, we assume that only the easily accessible features ($V$ and $X$) are available. In such case, a biased model (Figure 6(b)) is possibly constructed in which considers that $X$ directly influences $Y$. However, it is actually $W$ that contributes to $Y$, inferring that $X$ to $Y$ is an indirect path as shown in Figure 6(a). This leads to the concept of fair learning: sensitive terms such as nationalities, races, or genders ($X$) may affect the direct causes $W$ (which are not observable), and thus contribute to the target $Y$. Our objective is to avoid using such biased prior knowledge in Figure 6(b), and instead presume and identify the existence of $W$, as depicted in Figure 6(a). To this end, we generate $Z$ to represent $H(Y|X)$, which will cover part of the information of $V$ and $W$ (the unbiased part) and eliminate the biasing term $X$.

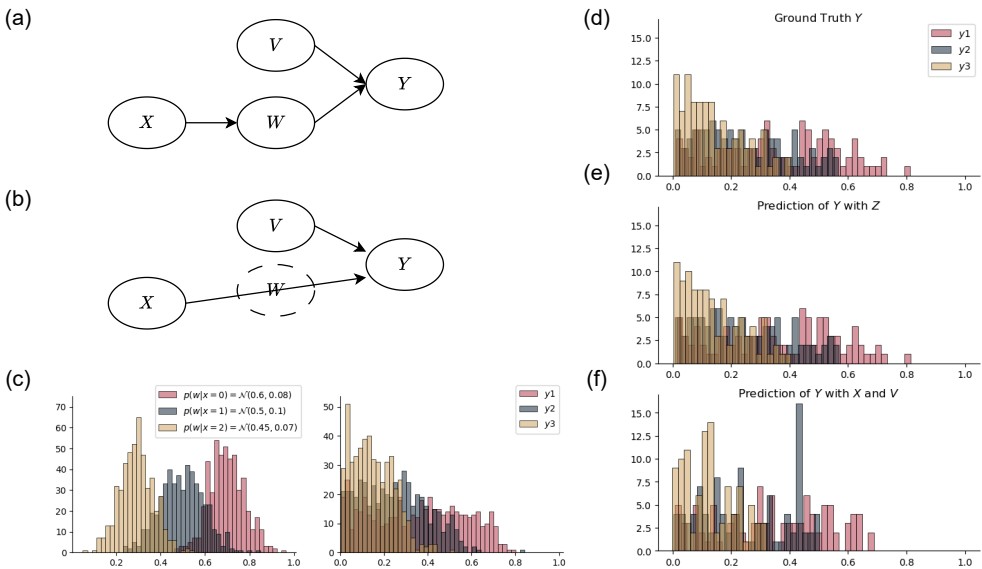

Figure 6: (a) The ground-truth relationship diagram between three features of scholars and their performances. (b) The biased relationship we may collect from the data. (c) The distribution of research training experience $W$ and academic performance $Y$ of scholars from countries $1, 2, 3$. (d) The ground truth $Y$ for test set, *i.i.d.* with the training set in (c). (e) Test prediction of $Y$ using generated $Z$. (f) Test prediction using biased features $X$ and $V$.

Table 2 demonstrates that the generated representation $Z$ contains the information about $H(Y|X)$ in this example. In Figure 6(f), we observe that the predictions of $Y$ using $\{X, V\}$ are biased, especially for researchers from country $x = 1$, which are skewed towards lower values. In contrast, predictions of $Y$ using $Z$ do not display such biased distributions.

We must notice that the task here is not prediction, as we need the target $Y$ itself as an input for generating $Z$, which poses a significant information leakage. Our focus in this section is just per-

Table 2: Experiment results: Information quantities among $X$, $Y$, and the generated $Z$.

| $I(X;Y)$ | $H(Y)$ | $H(Y|X)$ | $I(Z;Y)$ | $I(Z;X)$ | $I(Z,X;Y)$ | $I(Z;Y|X)$ |
|---|---|---|---|---|---|---|
| 0.25 | 3.87 | 3.62 | 0.19 | 2.98 | 3.01 | 2.76 |

forming Information Subtraction on $Y$ and illustrating how to generate unbiased features. In the next section, we will apply Information Subtraction to perform fair prediction tasks on real dataset. We should do it on the provided factor variables of $Y$, instead of $Y$ itself, to obtain a set of unbiased features. The generated features can then be used to predict $Y$ to avoid information leakage.

## 5.4 REAL CASE FOR FAIR LEARNING: ADULT INCOME DATA

In this section, we apply our framework to perform fair predictions on a common real dataset (Becker & Kohavi, 2024). The census data of 47621 individuals are collected, including their race, marital status, gender, and occupations. The goal is to use these attributes to predict whether the individuals have annual incomes over $50k$. In conventional fair learning studies, gender is considered as a protected variable, and the prediction should remain invariant over protected variable.

Denote $C$ as the protected variable, gender, $Y$ as the target variable, the income indicator, and $X$ as all other variables besides $C$ and $Y$. When $X$ is used as input to predict $Y$, and given that $X$ and $C$ are correlated, the information about $C$ within $X$ can result in biased estimations. Hence, $X$ is considered a set of biased features. We apply Information Subtraction to generate a set of unbiased features $Z$, which represent $H(X|C)$. This process retains most of the information from $X$ while eliminating the information of $C$. The prediction procedure is outlined in Algorithm D.1.

The experiment results of generating representations $Z$ for $H(X|C)$ are presented in Table 3 and 4.

Table 3: Experiment results: information quantities among $X$, $C$, and the generated $Z$.

| $I(S;G)$ | $H(G)$ | $H(G|S)$ | $I(Z;G)$ | $I(Z;S)$ | $I(Z,S;G)$ | $I(Z;G|S)$ |
|---|---|---|---|---|---|---|
| 0.62 | 6.86 | 6.24 | 3.25 | 0.02 | 4.17 | 3.55 |

Table 4: Experiment results: accuracy and group fairness metric of prediction $\hat{Y}$ using $\{X, C\}$, $X$ and $Z$.

| | $\hat{Y}_{\{X,C\}}$ | $\hat{Y}_X$ | $\hat{Y}_Z$ |
|---|---|---|---|
| Accuracy $\uparrow$ | 0.812 | **0.814** | 0.804 |
| BA $\uparrow$ | 0.641 | 0.628 | **0.810** |
| $Gap_C^{RMS} \downarrow$ | 0.081 | 0.077 | **0.033** |
| $Gap_C^{max} \downarrow$ | 0.094 | 0.091 | **0.037** |

We use two metrics of group fairness, balanced accuracy (BA) and Gap, to demonstrate the efficacy of our framework, provided in Appendix F. The results in Table 4 indicate that using $Z$ as an alternative of $X$ achieves significantly improved group fairness (higher BA and lower Gap). This suggests that Information Subtraction is capable of generating effective unbiased features $Z$.

## 5.5 REAL CASE FOR DOMAIN GENERALIZATION: COVER TYPE DATA

In the fair learning example, the data is collected from several domains, such as 'male' and 'female'. Both the training and testing sets contain data from these two domains, and it is assumed that the data in both sets is independently and identically distributed (i.i.d.). However, there are scenarios where the testing set is consisted of data that is out-of-distribution from the training set. We might train our models on data from multiple domains present in the training set, but subsequently test them on data from other domains. This challenge is known as domain generalization (Wang et al., 2022; Zhou et al., 2022).

For this purpose, we select the cover type dataset (Blackard, 1998), which includes features of $30 \times 30$ meter cells, such as elevation, aspect, slope, hillshade, and soil-type, across four distinct regions within the Roosevelt National Forest of Northern Colorado. Based on these features $X$, the task is to predict the predominant type of plant cover $Y$ in these cells. The label indicating the different domain areas $C$ is provided. The training set is composed of samples from three regions with higher elevation, while the testing set includes samples form the lowest region.

In this case, we have to estimate landscape testing samples that does not appear in the training set, and their distributions could be significantly different. Directly applying a model trained on the original features $X$ (which is correlated to $C$) could result in severe over-fitting. Our architecture is designed to generate representation $Z$ that excludes the domain-specific information (Zhu et al., 2021). By representing $H(X|C)$ with $Z$, the model focuses on learning general or universal features that are consistent across different regions. The standard metrics for domain generalization include the average-case and worst-case performance across different testing domains. In this instance, we have three training domains and a single testing domain. Therefore, we present the accuracy of predictions for the test domain using combinations of features $X, C, Z$ in the following table:

Table 5: Experiment results: accuracy of prediction $\hat{Y}$ using $\{X, C\}$, $X$, $Z$, and $\{X, Z\}$.

| | $\hat{Y}_{\{X,C\}}$ | $\hat{Y}_X$ | $\hat{Y}_Z$ | $\hat{Y}_{\{X,Z\}}$ |
|---|---|---|---|---|
| Accuracy | 0.590 | 0.566 | 0.483 | **0.598** |

After eliminating the information of $C$, $Z$ is expected to capture some universal features that are relevant for predicting $Y$. However, in this instance, $\hat{Y}_Z$ does not yield more accurate predictions than $\hat{Y}_X$. This suggests that regional information is more essential for predicting $Y$ than the universal representation, and we may not achieve domain generalization by eliminating the regional information in predictions.

Nevertheless, the universal features represented by $Z$ represent are still valuable in this scenario. We observe that $\hat{Y}_{X,Z}$, which includes both the regional information from $X$ and the universal features from $Z$, achieves the highest predictive accuracy. Notice that the universal features within $Z$, derived from Information Subtraction, cannot provide any new information. This is an implication of 'less is more': including $Z$, which consists of a portion of $X$'s information, drives the model away from the gradient descent trajectory of $\hat{Y}_X$ and reaches a better local optimum. Without introducing new information during training, this approach enhances the model's adaptability to out-of-distribution domains.

## 6 CONCLUSION

In this work, we have highlighted the significance of conditional representation learning. By capturing the relationships and effects between variables, our results demonstrate the potential applications of these representations in fair learning and domain generalization. We introduce a generative-based architecture for Information Subtraction, which is designed to generate representations that retain information about the target while eliminating that of the conditional variables.

We demonstrate that the resulting representations contain semantic features of the target that the conditional variables do not provide. Moreover, the unbiased representation has shown advanced fairness in both synthetic and real-world fair learning experiments. Finally, we demonstrate that combining factors $X$, which carry the domain-specific information, and representations $Z$, which carry universal features, leads to improved prediction for unseen out-of-distribution data.

Many future improvements can be made regarding this topic. Firstly, we note that the current overall objective is a simple weighted average of two information terms, and our architecture is not specifically designed for disentangling information. A more specialized architecture could achieve a better representation $Z$ with higher $I(X; Z, C)$ and lower $(Z; C)$. Additionally, in our domain generalization experiments, we simply put together $X$ and $Z$ as the input for estimating $Y$. An advanced integration of $X$ and $Z$ could lead to more satisfying predictions in future works.

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

# A   APPENDIX: IMPLEMENTATION DETAILS FOR PART 5.1 REPRESENTATION RELATIONSHIPS

The dynamics of the Lotka–Volterra model for the four species is sampled with discrete derivatives:

$$W(t + \Delta t) = W(t) + \Delta W(t) = W(t) + \frac{1}{\Delta t} W(t)(-a_0 + a_1 S(t) + a_2 R(t))$$

$$S(t + \Delta t) = S(t) + \Delta S(t) = S(t) + \frac{1}{\Delta t} S(t)(b_0 - b_1 W(t) + b_2 G(t))$$

$$R(t + \Delta t) = R(t) + \Delta R(t) = R(t) + \frac{1}{\Delta t} R(t)(c_0 - c_1 W(t) + c_2 G(t)) \tag{A.1}$$

$$G(t + \Delta t) = G(t) + \Delta G(t) = G(t) + \frac{1}{\Delta t} G(t)(d_0 - d_1 S(t) - d_2 R(t)),$$

where $W(t)$, $S(t)$, $R(t)$, and $G(t)$ represent the population of wolves, sheep, rabbits, and grass at time $t$, respectively.

To avoid divergent dynamics, we set the parameters and initials as follows:

$$W(0) = 9, \ R(0) = 10, S(0) = 10, \ G(0) = 100$$
$$a_0 = 9, a_1 = 0.3, a_2 = 0.1,$$
$$b_0 = 2, b_1 = 0.2, b_2 = 0.6,$$
$$c_0 = 3, c_1 = 0.2, c_2 = 0.8, \tag{A.2}$$
$$d_0 = 23, d_1 = 0.6, d_2 = 0.3,$$
$$\Delta t = 800$$

1500 samples are generated. While the data are generated from the synthetic distribution, the training and testing sets are guaranteed *i.i.d.* Hence, the training and testing sets are identical in this experiment. Other experiment details are listed in the following table:

Table A.1: Implementation details

| | |
|---|---|
| Dimension of $Z$ | 10 |
| Learning rate of the Generator | 1e-4 |
| Learning rate of the Discriminators | 5e-4 |
| Number of hidden layers in the Generator | 2 |
| Number of hidden layers in the Discriminators | 2 |
| Dimension of hidden layers in the Generator | 1000 |
| Dimension of hidden layers in the Discriminator | 1000 |

## B Appendix: Implementation Details for Part 5.2 Information Subtraction

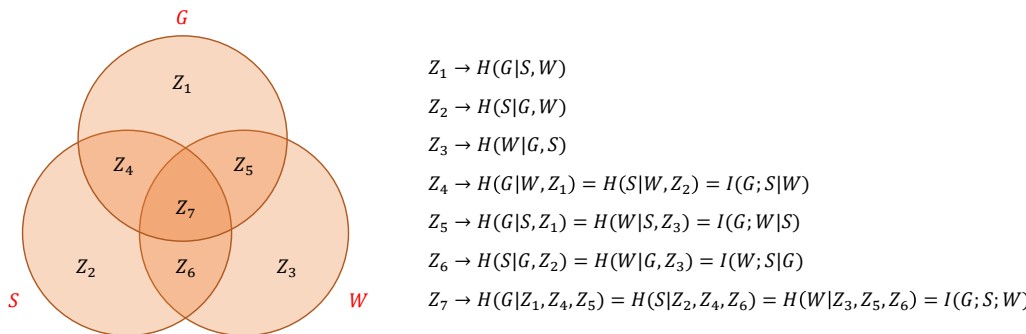

$$Z_1 \rightarrow H(G|S,W)$$
$$Z_2 \rightarrow H(S|G,W)$$
$$Z_3 \rightarrow H(W|G,S)$$
$$Z_4 \rightarrow H(G|W,Z_1) = H(S|W,Z_2) = I(G;S|W)$$
$$Z_5 \rightarrow H(G|S,Z_1) = H(W|S,Z_3) = I(G;W|S)$$
$$Z_6 \rightarrow H(S|G,Z_2) = H(W|G,Z_3) = I(W;S|G)$$
$$Z_7 \rightarrow H(G|Z_1,Z_4,Z_5) = H(S|Z_2,Z_4,Z_6) = H(W|Z_3,Z_5,Z_6) = I(G;S;W)$$

Figure B.1: Information Subtraction.

We use the variables $G$, $S$, and $W$ in the synthetic case in Appendix A for the demonstration of how Information Subtraction can obtain the representation of conditional entropy, mutual information, and conditional mutual information in Figure B.1.

First, we generate $Z_1$, which is the representation of conditional entropy $H(G|S,W)$. $Z_2$ and $Z_3$ can be achieved in similar ways. After we obtain $Z_1$, we can again apply the architecture of Information Subtraction to obtain $Z_4$ and $Z_5$ that represent $H(G|W,Z_1)$ and $H(G|S,Z_1)$ These two sectors correspond to the representation of conditional mutual information $I(G;S|W)$ and $I(G;W|S)$. Finally, we can further apply Information Subtraction to obtain $Z_7$ from $Z_1$, $Z_4$, and $Z_5$, which is the representation of trivariate mutual information $I(G;S;W)$. All these operations follow the same objective in Equation 6 with respective targets and condition variables.

1500 samples are generated. While the data are generated from the synthetic distribution, the training and testing sets are guaranteed *i.i.d*. Hence, the training and testing sets are identical in this experiment.

Table B.1: Implementation details

| | |
|---|---|
| Dimension of $Z_1$, $Z_4$, $Z_5$, $Z_7$ | 10 |
| Learning rate of the Generator | 1e-4 |
| Learning rate of the Discriminators | 5e-4 |
| Number of hidden layers in the Generator | 2 |
| Number of hidden layers in the Discriminators | 2 |
| Dimension of hidden layers in the Generator | 1000 |
| Dimension of hidden layers in the Discriminators | 1000 |

Table B.2: Experiment results: Information quantities among $G$, $S$, $W$, and the generated $Z_1$, $Z_4$, $Z_5$, $Z_7$.

| $H(G)$ | 3.32 | $H(S)$ | 3.37 | $H(W)$ | 4.75 |
|---|---|---|---|---|---|
| $I(G;S)$ | 0.93 | $I(S;W)$ | 0.96 | $I(W;G)$ | 0.77 |

| | | | | | |
|---|---|---|---|---|---|
| $I(Z_1;G\mid S,W)$ | 0.08 | $I(Z_1;S,W)$ | 1.14 | $I(G;S,W)$ | 2.20 |
| $I(Z_4;G\mid W,Z_1)$ | 0.25 | $I(Z_4;W,Z_1)$ | 1.26 | $I(G;W,Z_1)$ | 1.87 |
| $I(Z_5;G\mid S,Z_1)$ | 1.22 | $I(Z_5;S,Z_1)$ | 1.77 | $I(G;S,Z_1)$ | 1.72 |
| $I(Z_7;G\mid Z_1,Z_4,Z_5)$ | 0.00 | $I(Z_7;Z_1,Z_4,Z_5)$ | 1.29 | $I(G;Z_1,Z_4,Z_5)$ | 2.98 |

As shown in the second columns of Table B.2, we observe that $Z_1$, $Z_4$, and $Z_5$ effectively capture the desired conditional entropy. However, this framework shows ineffectiveness as $Z_1$, $Z_4$, $Z_5$,

and $Z_7$ still retain some information about the conditional variables that should be eliminated (the third columns). In other words, the framework cannot effectively disentangle and represent the information within each individual sector. This inefficiency can be attributed to several reasons.

Firstly, the variables in the synthetic case may not necessarily contain significant uncertainties, making it challenging for the architecture to learn to disentangle even a small amount of information between variables. Secondly, although MINE and SMILE can provide stable estimations of information, they are less precise when dealing with small information quantities. Thirdly, it is challenging to generate independent $Z_1$, $Z_4$, $Z_5$, and $Z_7$ from the same input $G$. Besides, independent variables do not necessarily imply zero estimation of the mutual information between them by MINE.

As stated in the conclusion, an architecture capable of more precise estimation of the information terms will generate better representations with higher $I(G; C, Z)$ and lower $I(C; Z)$, where $C$ denotes the conditional variables. An ideal architecture would address all the aforementioned problems: it would ultimately generate independent representations that disentangle the signals and encode different sectors within the Venn Diagram.

## C  APPENDIX: IMPLEMENTATION DETAILS FOR PART 5.3 SYNTHETIC CASE FOR FAIR LEARNING

1500 samples are generated. While the data are generated from the synthetic distribution, the training and testing sets are guaranteed *i.i.d.* Hence, the training and testing sets are identical in this experiment. The experiment details are listed in the following table:

Table C.1: Implementation details

| | |
|---|---|
| Dimension of $Z_1$, $Z_4$, $Z_5$, $Z_7$ | 10 |
| Learning rate of the Generator | 1e-4 |
| Learning rate of the Discriminators | 5e-4 |
| Number of hidden layers in the Generator | 2 |
| Number of hidden layers in the Discriminators | 2 |
| Dimension of hidden layers in the Generator | 1000 |
| Dimension of hidden layers in the Discriminators | 1000 |

# D APPENDIX: IMPLEMENTATION DETAILS FOR PART 5.4 REAL CASE FOR FAIR LEARNING

32621 samples are included in the datasets, with 15000 training samples and 17621 testing samples. After training, the results in Table 3 are evaluated with the training samples, and the results in Table 4 are evaluated with the testing samples. The experiment details are listed in the following table:

Table D.1: Implementation details

| | |
|---|---|
| Dimension of $X$ | 105 |
| Dimension of $C$ | 1 |
| Dimension of $Y$ | 1 |
| Dimension of $Z$ | 100 |
| Learning rate of the Generator | 1e-4 |
| Learning rate of the Discriminators | 5e-4 |
| Learning rate of all Estimators | 1e-4 |
| Number of hidden layers in the Generator | 2 |
| Number of hidden layers in the Discriminators | 2 |
| Number of hidden layers in all Estimators | 2 |
| Dimension of hidden layers in the Generator | 5000 |
| Dimension of hidden layers in the Discriminators | 5000 |
| Dimension of hidden layers in all Estimators | 5000 |

In Section 5.3, we mention that we should generate unbiased features from the factor variables, instead of the target itself, to avoid information leakage in predictions. Algorithm D.1 provides the corresponding pseudo-code for the unbiased predictions.

---

**Algorithm D.1** Performing Prediction By Generating Unbiased Features

---

**Input:** biased factor variables $X$, conditional variables $C$, target $Y$, $n_1 < n_2, n_3, n_4$;
**Output:** $Z, N_E$;
1: Initialize neural networks: generator $N_A$, reconstructor $N_B$, discriminator $N_C$ for $I(Y; X, Z)$, discriminator $N_D$ for $I(X; Z)$, estimator $N_E$ for $\hat{Y}$;
2: **for** $i = 1, 2, \cdots, n_2$ **do**
3:     Sample batch $x, c$;
4:     $z \leftarrow N_A(x)$;
5:     **if** $i < n_1$ **then**
6:         Update $N_A$ w.r.t. minimizing $l_1 = \|N_B(z) - x\|$;
7:     **else**
8:         Update $N_A$ w.r.t. minimizing $l_2 \leftarrow N_D(c, z) - N_C(x, c, z) = I(C; Z) - I(X; C, Z)$;
9:     **end if**
10:    **for** $j = 1, 2, \cdots, n_3$ **do**
11:       Sample batch $x, c$;
12:       $z \leftarrow N_A(y)$;
13:       Update $N_C$ w.r.t. minimizing $l_3 \leftarrow -N_C(x, y, z) = -I(Y; X, Z)$;
14:       Update $N_D$ w.r.t. minimizing $l_4 \leftarrow -N_D(x, z) = -I(X; Z)$;
15:    **end for**
16: **end for**
17: **for** $k = 1, 2, \cdots, n_4$ **do**
18:    Sample batch $x, y$;
19:    $z \leftarrow N_A(x)$;
20:    $\hat{y} \leftarrow N_E(z)$
21:    Update $N_E$ w.r.t. minimizing $l_1 = \|\hat{y} - y\|$;
22: **end for**

---

# E APPENDIX: IMPLEMENTATION DETAILS FOR PART 5.5 REAL CASE FOR DOMAIN GENERALIZATION

The training set is consisted of 130000 samples from 3 domains, and the testing set is consisted of 29884 samples from 1 domain. The results in Table 5 is evaluated with the testing samples. The experiment details are listed in the following table. The prediction procedure follows Algorithm D.1

Table E.1: Implementation details

| | |
|---|---|
| Dimension of $X$ | 50 |
| Dimension of $C$ | 1 |
| Dimension of $Y$ | 1 |
| Dimension of $Z$ | 50 |
| Learning rate of the Generator | 1e-4 |
| Learning rate of the Discriminators | 5e-4 |
| Learning rate of all Estimators | 1e-4 |
| Number of hidden layers in the Generator | 2 |
| Number of hidden layers in the Discriminators | 2 |
| Number of hidden layers in all Estimators | 2 |
| Dimension of hidden layers in the Generator | 2500 |
| Dimension of hidden layers in the Discriminators | 2500 |
| Dimension of hidden layers in all Estimators | 2500 |

# F APPENDIX: GROUP FAIRNESS METRICS

Denote $C$ as the protected variable, gender, $Y$ as the target variable, income indicator, $\hat{Y}$ as the prediction of $Y$, and $X$ be the other attributes besides $C$ and $Y$. Here, we assume that $C$ is a binary variable $c = 0, 1$, and $Y$ is a categorical variable, $y = \{1, 2, ..., n\}$.

For a specific value $c$ and $y$, we define

$$TPR_{c,y} = p(\hat{Y} = y | C = c, Y = y) \tag{F.1}$$

Then, we can define

$$GAP_{C,y} = TPR_{0,y} - TPR_{1,y} \tag{F.2}$$

For all the value $y$ of $Y$, we define

$$GAP_C^{RMS} = \sqrt{\frac{1}{n} \sum_{y \in Y} Gap_{C,y}^2} \tag{F.3}$$

$$GAP_C^{max} = argmax_{y \in Y} |Gap_{C,y}| \tag{F.4}$$

Another metric, balanced accurate (BA), is defined as

$$BA = \frac{1}{n} \sum_{y \in Y} p(\hat{Y} = \hat{y} | Y = y) \tag{F.5}$$

# G APPENDIX: SENSITIVE ANALYSIS OF HYPER-PARAMETER $\lambda$

Figure G.1 shows the sensitive analysis results using different hyper-parameter $\lambda$ from Equation 10 in Section 5.1 and 5.3. $\lambda$ balances the trade-off between the two objectives: maximizing $I(Y; X, Z)$ and minimizing $I(Y; Z)$. Different $\lambda$ leads to different locations $Z$ places on the information plane (Zhao et al., 2022).

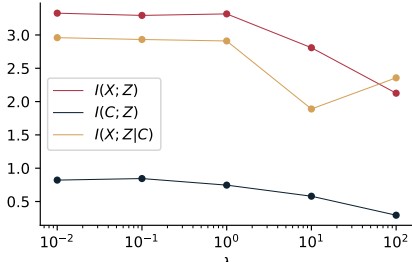 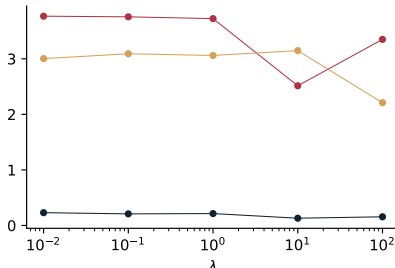

Figure G.1: Sensitivity analysis of different $\lambda$ in Section 5.1 and 5.3.

