# OpenReview forum: "Information Subtraction: Learning Representations for Conditional Entropy"
_ICLR.cc/2025/Conference — Submitted to ICLR 2025_

### Official Review · Reviewer_zrA2 · 2024-10-30

**Soundness:** 2
**Presentation:** 1
**Contribution:** 2
**Rating:** 3
**Confidence:** 3

**Summary:**

This paper introduces the Information Subtraction framework, which addresses conditional representation learning for continuous variables. It employs a generative architecture with a generator neural network and two discriminators to stabilize information term estimations. The generator's objective is to maximize information from one discriminator while minimizing it from the other, effectively capturing semantic features of conditional entropy and enhancing fair learning by removing sensitive information.
The authors highlight the significance of conditional representation learning and demonstrate the framework's capacity to decompose signals and produce unbiased representations. Experimental results show that the proposed approach improves fairness in both synthetic and real-world contexts and enhances domain generalization by combining domain-specific factors with universal representations.

**Strengths:**

The paper effectively outlines the problem from a methodological standpoint, framing it as a well-defined optimization issue that is clear from a mathematical perspective. It provides a comprehensive overview of related work and their limitations, emphasizing the necessity for a representation learning method that can eliminate information pertaining to continuous sensitive variables. The primary objective is to elucidate the unique effects among variables. Additionally, the inclusion of real-world experiments highlights the paper's contributions to the current research landscape in fair learning and domain generalization applications.

**Weaknesses:**

The article presents challenges in terms of readability, as the mathematical loss functions being minimized in practice are not adequately described. The discussion of the quantities to be optimized tends to remain at a high level. Additionally, the architecture description is introduced late in the paper and lacks detail; a more concrete schematic representation with specific input types (e.g., images, tabular data) and detailed analytical loss expressions would be beneficial.

Moreover, the existing literature on debiasing appears to be quite extensive regarding the elimination of sensitive information from continuous variables [A], [B]. I found it difficult to discern how this work connects to those prior studies.

Finally, the experimental section seems relatively weak in terms of the number of experiments and datasets utilized. Including a straightforward debiasing or fair learning experiment in a real-world context, such as healthcare applications or scenarios involving ethnic biases, along with qualitative visual explanations, would enhance the overall quality of the article.

[A] Unbiased Supervised Contrastive Learning, C.A. Barbano et al. ICLR 2023. https://arxiv.org/abs/2211.05568
[B] CLUB: A Contrastive Log-ratio Upper Bound of Mutual Information, P. Cheng et al, ICML 2020. https://arxiv.org/abs/2006.12013

**Questions:**

Is it necessary to assume that your inputs or latent representations follow a specific distribution (e.g., Gaussian, von Mises-Fisher) to derive your loss functions?

Additionally, the authors mention "Based on our previous work (?)" at one point. It is important to note that ICLR requires authors to cite their own work as they would cite others'. In this case, the authors should state: "Based on the previous work [x]." This would allow readers to locate and review the referenced paper.

---

> ### Author Response · Authors · 2024-11-27
> **Response for the weaknesses 1, 2, 3**
>
> First of all, please allow us to express our gratitude for your valuable suggestions. Here are the responds for the weaknesses 1, 2, 3.
>
> Q1: The article presents challenges in terms of readability, as the mathematical loss functions being minimized in practice are not adequately described. The discussion of the quantities to be optimized tends to remain at a high level.
>
> A1: We greatly appreciate your suggestions. We have made revisions to Equation (2, 4, 5, 6, 10) and Algorithm (1). If you find any issues with the revised version, we would be grateful if you could specify the detailed concerns. This feedback would enable us to address these points and enhance the clarity and presentation of our manuscript.
>
> Q2: Additionally, the architecture description is introduced late in the paper and lacks detail; a more concrete schematic representation with specific input types (e.g., images, tabular data) and detailed analytical loss expressions would be beneficial.
>
> A2: We have revised Figure (3) and its captions to provide a detailed explanation. We do not specify input types simply because we have not put a restriction on them. We kindly request you to review the updated version. If you find any issues with the revised version, we welcome further discussion with you.
>
> Regarding the placement of the architecture section, we are open to your guidance on whether it should precede the related work section, or if you believe it would be more appropriate to position the related works elsewhere.
>
> Q3. Moreover, the existing literature on debiasing appears to be quite extensive regarding the elimination of sensitive information from continuous variables [A], [B]. I found it difficult to discern how this work connects to those prior studies.
> [A] Unbiased Supervised Contrastive Learning, C.A. Barbano et al. ICLR 2023. https://arxiv.org/abs/2211.05568
> [B] CLUB: A Contrastive Log-ratio Upper Bound of Mutual Information, P. Cheng et al, ICML 2020. https://arxiv.org/abs/2006.12013
>
> A3: We sincerely appreciate the literature recommendations provided. We agree that works [A] and [B] are closely related to our work. The paper [A] belongs to supervised contrastive learning, which requires both a target label and a conditional label to be provided. However, the problem addressed in our manuscript is situated within a self-supervised setting, implying that our training loss function does not encompass a target label, but is instead generating a representation that filters the conditional label's information from the input. Consequently, it is not apt to serve as a baseline for comparison with our work, and we find it hard to discuss it in our work.
>
> [B] is a deep learning-based mutual information estimator. In this estimator, the input is a fixed distribution of X, Y, and Z, and the estimator is trained using samples from this distribution, with the output being the information estimation. Our representation learner, however, consists of two parts: an encoder and an information estimator. The input to the information estimator in our representation learner is not fixed, as the distribution of Z is not constant during the training process. We have experimented with various estimators in our preliminary experiments, including MINE, CCMI, and CLUB. We did make a lot of attempts on CLUB, as it is the only one that estimates the upper bound, while the others estimate the lower bound, making it particularly useful for our information minimization part. Unfortunately, we found that only MINE worked, which is why we did not choose CCMI and CLUB. Still, we agree that mentioning other methods in Line 226 could be beneficial.
>
> In accordance with the suggestions from you and other reviewers, we are currently in the process of supplementing two contrastive-based baselines, one for discrete scenarios [1] and another for continuous scenarios [2]. The code for these has been successfully replicated; however, the results significantly underperform compared to our method. We are currently investigating whether this is due to issues with experimental parameters or if their contrastive-based frameworks are ineffective in carrying conditional mutual information within our experimental context. Therefore, in this rebuttal revised version, we have not included yet the comparative experimental results. Nevertheless, we commit to incorporating these findings in the final camera-ready version.
> [1] Martin Q Ma, Yao-Hung Hubert Tsai, Paul Pu Liang, Han Zhao, Kun Zhang, Ruslan Salakhutdinov, and Louis-Philippe Morency. Conditional contrastive learning for improving fairness in self-supervised learning. arXiv preprint arXiv:2106.02866, 2021.
> [2] Yao-Hung Hubert Tsai, Tianqin Li, Martin Q Ma, Han Zhao, Kun Zhang, Louis-Philippe Morency, and Ruslan Salakhutdinov. Conditional contrastive learning with kernel. In International Conference on Learning Representations. 2022.

---

> ### Author Response · Authors · 2024-11-27
> **Response for Weakness 4 and Questions 1,2**
>
> Here are the responds for the WEAKNESS 4.
>
> Q4: Finally, the experimental section seems relatively weak in terms of the number of experiments and datasets utilized. Including a straightforward debiasing or fair learning experiment in a real-world context, such as healthcare applications or scenarios involving ethnic biases, along with qualitative visual explanations, would enhance the overall quality of the article.
>
> A4: We wish to note that the adult dataset is one of the most commonly used examples in fair learning. Besides, we believe that the plant cover dataset used in Section 5.5 is an excellent example of a domain generalization problem. The data distribution within this dataset aligns with actual geographical features and ecological distributions, providing an intuitive explanation of the relationship between input features and domains.
>
> We agree that qualitative visual explanations could be very helpful for better illustrating our results. We have not yet come up with a solution, and we are very interested to discuss with your further suggestion.
>
> Here are the responds for the QUESTIONS.
>
> Q1: Is it necessary to assume that your inputs or latent representations follow a specific distribution (e.g., Gaussian, von Mises-Fisher) to derive your loss functions?
>
> A1: This is not a requirement, and it is, in fact, one of the strengths of our framework. Compared to structures like VAE, our architecture imposes no distributional restrictions on the input, representation, or output. However, this is not the unique contribution of this paper, hence we did not highlight it. Nevertheless, it is indeed worth mentioning to alleviate any potential confusion.
>
> Q2: Additionally, the authors mention "Based on our previous work (?)" at one point. It is important to note that ICLR requires authors to cite their own work as they would cite others'. In this case, the authors should state: "Based on the previous work [x]." This would allow readers to locate and review the referenced paper.
>
> A2: Thank you for clarifying the rules. We originally thought that citing our own paper in the anonymous paper will lead to desk rejection. We apologize for the mistake, but we decided not to put it on the revised version now, or it will be too obvious for the reviewers to notice our identity. The reference will be available in the camera-ready version.
>
> Thank you again for your time and your insightful questions.

---

### Official Review · Reviewer_Q4Lr · 2024-10-30

**Soundness:** 2
**Presentation:** 2
**Contribution:** 2
**Rating:** 5
**Confidence:** 3

**Summary:**

The authors introduce a framework for representing arbitrary information components between continuous variables using Information Subtraction. Essentially a generator network is trained to generate a latent representation $Z$ which captures the mutual entropy of target viable $Y$ and conditional variable $X$, without carrying information about $X$ itself.
To achieve this, they use two discriminator networks $A$ and $B$. While $A$ estimates $I(Z,X;Y)$, $B$ estimates $I(Z;X)$. The Objective $I(Z,X;Y) - I(Z;X)$ is back propagated to the generator network.

The authors test their method on two synthetic scenarios and on fair learning and domain generalisation.

**Strengths:**

The paper is well written in general. The challenge being tackled is of interest for part of ICLR's audience. While maybe not novel – I am not an expert on the related work – the approach of using two discriminator networks to estimate $I(Z,X;Y)$ and $I(Z;X)$ seems reasonable to me.

**Weaknesses:**

**Related Work Section**:

The authors write: "While we share similar architectures with these works, their structures are not designed for conditional representations."
It is unclear to me how large the contribution of this paper is. Is the proposed architecture only a slight modification of existing work? Or would a slight modification of existing work suffice to reach the same goal as the authors propose? If so, why is there no comparison to those in the experimental section?

**Experimental Section**:

The section lacks comparability to prior work. As I understand it, it stands very much isolated and it's hard for me to estimate the significance of  the contribution the authors have made. If existing models cannot be applied for comparison, I'd still expect the authors to come up with other, simpler, baseline architectures against which to compare.

It is unclear to me whether the reported values come from train, validation or test splits. The lack of standard deviation (suggesting no cross validation was used) makes it hard to estimate the significance of the results. Additionally, the chosen "real-world" datasets seem very simple to me.

Overall, unfortunately, the experimental section does not convince me.

**Questions:**

The *Weaknesses* section outlines the questions and concerns I have.

---

> ### Author Response · Authors · 2024-11-27
> **Response for the first weakness**
>
> First of all, please allow us to express our gratitude for your valuable suggestions. Here are the responds for the first weakness.
>
> Q1: Related Work Section:The authors write: "While we share similar architectures with these works, their structures are not designed for conditional representations." It is unclear to me how large the contribution of this paper is. Is the proposed architecture only a slight modification of existing work? Or would a slight modification of existing work suffice to reach the same goal as the authors propose? If so, why is there no comparison to those in the experimental section?
>
> A1: Firstly, our primary contribution lies in the introduction of the concept of information subtraction and demonstrating how it theoretically represents various information sector elements within a Venn diagram. We have also showcased the significant applications of information subtraction, including its role in fair learning and domain generalization.
>
> To realize this concept, we have designed an architecture aimed at achieving this goal. Our generator-discriminator architecture consists of two broad modules, and we have not provided detailed specifications or discussions on which types of neural networks should be employed for each component, because they could be very flexible. For instance, while we mention in the paper that the generator could be an RNN, CNN, or ResNet, we have only utilized FNN. Similarly, in our revised version, we indicate that the discriminator could be MINE, CLUB, or CCMI, but our experiments have revealed that only MINE yields satisfactory results.
>
> Regarding the similarity to related works, it is important to clarify that many related works employ a generator-discriminator architecture, but the specific implementation of our FNN+MINE structure, the optimization logic, and the optimization objectives are uniquely ours. Other methods used in the related works, such as contrastive-based and surprised-based approaches, are more or less inapplicable within the context of our information subtraction concept. These methods have their own set of challenges and future research trends, which further highlight the distinctiveness of our approach. To avoid any confusion, we think it will be more appropriate to the similarity of objectives instead of architecture in line 164.
>
> In accordance with the suggestions from you and other reviewers, we are currently in the process of supplementing two contrastive-based baselines, one for discrete scenarios [1] and another for continuous scenarios [2]. The code for these has been successfully replicated; however, the results significantly underperform compared to our method. We are currently investigating whether this is due to issues with experimental parameters or if their contrastive-based frameworks are ineffective in carrying conditional mutual information within our experimental context. Therefore, in this rebuttal revised version, we have not included yet the comparative experimental results. Nevertheless, we commit to incorporating these findings in the final camera-ready version.
>
> [1] Martin Q Ma, Yao-Hung Hubert Tsai, Paul Pu Liang, Han Zhao, Kun Zhang, Ruslan Salakhutdinov, and Louis-Philippe Morency. Conditional contrastive learning for improving fairness in self-supervised learning. arXiv preprint arXiv:2106.02866, 2021.
>
> [2] Yao-Hung Hubert Tsai, Tianqin Li, Martin Q Ma, Han Zhao, Kun Zhang, Louis-Philippe Morency, and Ruslan Salakhutdinov. Conditional contrastive learning with kernel. In International Conference on Learning Representations. 2022.

---

> ### Author Response · Authors · 2024-11-27
> **Response for the second weakness**
>
> Here are the responds for the second weakness.
>
> Q2: Experimental Section: The section lacks comparability to prior work. As I understand it, it stands very much isolated and it's hard for me to estimate the significance of the contribution the authors have made. If existing models cannot be applied for comparison, I'd still expect the authors to come up with other, simpler, baseline architectures against which to compare.
> It is unclear to me whether the reported values come from train, validation or test splits. The lack of standard deviation (suggesting no cross validation was used) makes it hard to estimate the significance of the results. Additionally, the chosen "real-world" datasets seem very simple to me.
> Overall, unfortunately, the experimental section does not convince me.
>
> A2: We fully agree that including comparative experiments can underscore the significance of our findings. As mentioned previously, we are currently in the process of supplementing our study with two contrastive-based baselines, which will be included in the final camera-ready version.
>
> In our work, different sections utilize distinct train/test splits. In the synthetic case (Sections 5.1 to 5.3), the data is generated from the same provided distribution. Therefore, there is no necessity for splitting, and the entire dataset serves as the training set. In contrast, in the real cases (Sections 5.4, 5.5), where empirical data are employed, the assumption of i.i.d. can hardly be sustained. Consequently, the data in these sections have been partitioned into training and testing sets. We have incorporated the corresponding explanation in Appendix A, B, C, D, and E, accordingly.
>
> We wish to note that the adult dataset is one of the most commonly used examples in fair learning. The main purpose of this paper is to propose a fundamental model, so we wish to demonstrate that our architecture can perform basic classification tasks effectively. We choose not to work on Computer Vision tasks, as conditional mutual information in the context of CV may not be straightforward to interpret. Besides, we believe that the plant cover dataset used in Section 5.5 is an excellent example of a domain generalization problem. The data distribution within this dataset aligns with actual geographical features and ecological distributions, providing an intuitive explanation of the relationship between input features and domains.
>
> Thank you again for your time and your insightful questions.

---

### Official Review · Reviewer_KedS · 2024-11-03

**Soundness:** 3
**Presentation:** 2
**Contribution:** 1
**Rating:** 3
**Confidence:** 3

**Summary:**

This paper proposes a framework called "Information Subtraction" for learning representation Z that maximizes conditional entropy H(Y|X) or, put in another way, maximizes the conditional mutual information (CMI) I(Z;Y|X). The method applies for continuous variables, which is harder than discrete variables. The proposed framework utilizes an approach similar to generative adversarial training where discriminators are used to maximize or minimize information terms. The authors evaluate the framework's performance on synthetic and real-world datasets, demonstrating its effectiveness in fair learning and domain generalization tasks.

**Strengths:**

The framework tackles a relatively under-explored and challenging problem of selectively maximizing and minimizing specific information components during representation learning. Previous works in the topic of CMI are more focusing on getting a good estimation, while this work is interested in leveraging CMI for representation learning.

**Weaknesses:**

**Lack of background of CMI estimators**: Previous works on estimating (conditional) mutual information are not discussed. In particular, [1] proposes similar framework where discriminators are used for the estimation. It's also unclear to me how the proposed method differs from the existing approaches for estimating conditional mutual information, e.g. MI-Diff+f-MINE in [1]. Are there any significant technical difficulties for turning an estimator into a representation learner? Would the classifier-based approach advocated by [1] results in better representation?

**Lack of baselines and ablation studies**: the experiments, synthetic or real, don't compare to any other methods. It's, therefore, not obvious how the experimental result should be interpreted. The experimental section would also benefit from ablation studies to analyze the contribution of different components of the architecture and the impact of hyperparameter choices.

[1] Mukherjee et al. CCMI : Classifier based Conditional Mutual Information Estimation.

**Questions:**

1. Are there any difficulties or details that need to pay attention to make the training scheme work? MINE has been proved to be difficult to tune.
2. In the paper, the generator takes Y as the input, could X also be given in the input? Would that change the result?
3. Line 168, reference missing. Why is it called an expanded architecture?
4. For the synthetic experiments, does the proposed method learns a better representation than the baselines such as contrastive-based approach?
5. Does the proposed method perform better in the downstream task than other approaches for the fairness and domain invariant learning setting?

---

> ### Author Response · Authors · 2024-11-27
> **Response for the weaknesses**
>
> First of all, please allow us to express our gratitude for your valuable suggestions. Here are the responds for the weaknesses.
>
> Q1: Lack of background of CMI estimators: Previous works on estimating (conditional) mutual information are not discussed. In particular, [1] proposes similar framework where discriminators are used for the estimation. It's also unclear to me how the proposed method differs from the existing approaches for estimating conditional mutual information, e.g. MI-Diff+f-MINE in [1]. Are there any significant technical difficulties for turning an estimator into a representation learner? Would the classifier-based approach advocated by [1] results in better representation?
> [1] Mukherjee et al. CCMI : Classifier based Conditional Mutual Information Estimation.
>
> A1: We would like to express our gratitude for the suggestions provided. The use of deep learning methods to estimate conditional mutual information is indeed relevant to our research. The reason we did not include this in the related works section is that we wished to focus on the aspect that our work belongs to representation learning. Generating representation for conditional mutual information is our central focus, and the CCMI approach you mentioned does not employ the inputs' representations while estimating conditional mutual information. Therefore, we did not include it in the main text of the related works; however, we agree that a short discussion can be placed in Line 226.
>
> Addressing your second point, we first clarify that the estimator is trained using samples from a fixed distribution of X, Y, and Z. In contrast, a representation learner consists of two components: an encoder and an information estimator. The input for the information estimator within the representation learner is not fixed, as the distribution of Z keeps optimizing during the training process, which poses a challenge for the estimator.
>
> This leads to your third point. In our preliminary experiments, we tested various information estimators, including MINE, CCMI, and CLUB, and we found that only MINE was effective. This is the reason we did not choose other estimators.
>
> Q2: Lack of baselines and ablation studies: the experiments, synthetic or real, don't compare to any other methods. It's, therefore, not obvious how the experimental result should be interpreted. The experimental section would also benefit from ablation studies to analyze the contribution of different components of the architecture and the impact of hyperparameter choices.
>
> A2: The primary focus of our current manuscript is on the theoretical framework, with the major contribution being the proposal of a representation for conditional entropy as an issue and concept. The emphasis of the experimental section is to demonstrate that the representation Z we generate performs better in fair learning and domain generalization compared to the original input X, rather than presenting an incremental or SOTA approach.
>
> Another reason we did not include too many baselines is that most of the related works do not address the representation of conditional entropy. However, as you rightly pointed out, comparisons would better illustrate the importance of our framework. Therefore, we are currently in the process of supplementing two contrastive-based baselines, one for discrete scenarios [1] and another for continuous scenarios [2]. The code for these has been successfully replicated; however, the results significantly underperform compared to our method. We are currently investigating whether this is due to issues with experimental parameters or if their contrastive-based frameworks are ineffective in carrying conditional mutual information within our experimental context. Therefore, in this rebuttal revised version, we have not included yet the comparative experimental results. Nevertheless, we commit to incorporating these findings in the final camera-ready version.
>
> Additionally, the hyperparameter sensitivity analysis you suggested is indeed crucial. We should incorporate a weight lambda into equation (10) and conduct a sensitivity analysis. The experimental results of sensitivity analysis have been included in Table G.1, G.2 of Appendix G.
>
> [1] Martin Q Ma, Yao-Hung Hubert Tsai, Paul Pu Liang, Han Zhao, Kun Zhang, Ruslan Salakhutdinov, and Louis-Philippe Morency. Conditional contrastive learning for improving fairness in self-supervised learning. arXiv preprint arXiv:2106.02866, 2021.
> [2] Yao-Hung Hubert Tsai, Tianqin Li, Martin Q Ma, Han Zhao, Kun Zhang, Louis-Philippe Morency, and Ruslan Salakhutdinov. Conditional contrastive learning with kernel. In International Conference on Learning Representations. 2022.

---

> ### Author Response · Authors · 2024-11-27
> **Response for the questions**
>
> Here are the responds for the questions.
>
> Q1: Are there any difficulties or details that need to pay attention to make the training scheme work? MINE has been proved to be difficult to tune.
>
> A1: You are entirely correct in your observation that MINE is well-known for its instability during the training process. This instability primarily stems from the susceptibility of the upper bound used by MINE to significant fluctuations due to minor parameter updates in the neural network. Additionally, simply reducing the learning rate to mitigate these fluctuations can lead to issues such as slow training. We should acknowledge that careful selection of parameters is required during training, which is almost a universal issue for all frameworks utilizing MINE, and is one of the tradeoffs for enjoying the simplicity of MINE. We have provided the learning rates and network sizes for all the experiments in Table A.1, B.1, C.1, D.1, and E.1.
>
> Q2: In the paper, the generator takes Y as the input, could X also be given in the input? Would that change the result?
>
> A2: Given our objective to eliminate the information of X from Y to obtain Z, we believe that incorporating X as an input of the generator may not be particularly beneficial, and could potentially lead to Z containing more information from X than desired. We understand your concern might stem from the possibility that when X is input into the generator, the generator might learn the characteristics of X and avoid outputting it. This is a plausible scenario for a black-box generator, and we can further explore in our future work.
>
> Q3: Line 168, reference missing. Why is it called an expanded architecture?
>
> A3: It is our previous work. To avoid violating the double-blinded review protocol, we temporarily concealed this paper during the review stage and included a placeholder. The main contribution of this previous work was the use of MINE as a discriminator, and our current submission significantly expands upon this foundation. The placeholder will be filled in the camera-ready version.
>
> Q4: For the synthetic experiments, does the proposed method learns a better representation than the baselines such as contrastive-based approach?
>
> A4: As we have mentioned previously, we are currently in the process of supplementing two contrastive-based baselines, one for discrete scenarios [1] and another for continuous scenarios [2]. We commit to incorporating these results in the final camera-ready version.
>
> [1] Martin Q Ma, Yao-Hung Hubert Tsai, Paul Pu Liang, Han Zhao, Kun Zhang, Ruslan Salakhutdinov, and Louis-Philippe Morency. Conditional contrastive learning for improving fairness in self-supervised learning. arXiv preprint arXiv:2106.02866, 2021.
> [2] Yao-Hung Hubert Tsai, Tianqin Li, Martin Q Ma, Han Zhao, Kun Zhang, Louis-Philippe Morency, and Ruslan Salakhutdinov. Conditional contrastive learning with kernel. In International Conference on Learning Representations. 2022.
>
> Q5: Does the proposed method perform better in the downstream task than other approaches for the fairness and domain invariant learning setting?
>
> A5: The objective of the experiments conducted in Sections 5.4 and 5.5 is to demonstrate that the generated Z provides more effective information than the original input features X and better accomplishes the task of fair and domain-invariant learning, thereby illustrating the effectiveness of information subtraction. Comparing with other approaches is not the focus of our experiments; however, we agree that in subsequent work, when we further explore the application of information subtraction in fair learning, we should conduct a thorough comparison with other relevant studies.
>
> Thank you again for your time and your insightful questions.

---

> > ### Comment · Reviewer_KedS · 2024-11-28
> >
> > Thank you for your response. The exploration of different conditional mutual information estimators is commendable, and including these results with an analysis of their limitations would significantly strengthen the justification for your proposed design. The addition of more baselines is also welcomed. However, in its current form, the paper requires further revisions to incorporate feedback from the reviewers. I encourage the authors to resubmit your work after addressing these points.

---

### Official Review · Reviewer_EVYs · 2024-11-04

**Soundness:** 1
**Presentation:** 1
**Contribution:** 1
**Rating:** 3
**Confidence:** 3

**Summary:**

The authors introduce a framework for learning representations, aimed at applications in fair learning and domain generalization.
The authors use powerful MINE-based estimators to learn representations that share minimal MI with given conditional variables.
Previous methods focused on discrete sensitive variables, while here the authors extend these approaches to continuous cases.

**Strengths:**

* The choice of MINE-based MI estimator to maximize/minimize MI terms is an excellent choice which allows the proposed method to scale to high-dimensional data.
* Extending previous work to include continuous variables is an important contribution.
* The proposed method might have potential for broad application due to the above points, and the fact that it is independent of the choice of architecture.

**Weaknesses:**

* Line 168: “Based on our previous work (?)” - this is a strong hint to the identity of the authors which breaks the double-blind regime of the reviewing process.
* The novelty might be very limited here. Specifically, the use of MINE-based methods might be the major novelty here.
* There is no discussion about the failing points of the proposed method. What happened when the condition variable X and the target Y are entangled in more complex ways? How will the learned representation Z be affected?
* The formulation and the presented Algorithm are not clear.

**Questions:**

* Did you try adding a hyper-parameter to one of the terms in the loss? Could that allow for finer control by the user on the learned representation?
* Did you have issues in training stability?
* Did you test the effectiveness of the proposed method on high dimensional data? How did the computational cost scale?

---

> ### Author Response · Authors · 2024-11-27
> **Response for the weakness**
>
> First of all, please allow us to express our gratitude for your valuable suggestions. Here are the responds for the weakness.
>
> Q1: Line 168: “Based on our previous work (?)” - this is a strong hint to the identity of the authors which breaks the double-blind regime of the reviewing process.
>
> A1: We do not consider that this approach disobeys the double-blind review process, as it does not disclose excessive information. We just want to put a placeholder here without directly showing our names in the anonymous manuscript. However, we do realize that such a description could allow reviewers who are familiar with our previous work to infer our identity during the review process. We apologize for this flaw.
>
> Q2: The novelty might be very limited here. Specifically, the use of MINE-based methods might be the major novelty here.
>
> A2: The use of MINE to generate representations is actually the major novelty of our previous research. In this manuscript, the major contribution lies in the introduction of Information Subtraction, which is a concept achieved by generating representation for conditional entropy. We have formulated the mathematical expressions necessary to achieve this objective and proposed a neural network architecture to implement it. Furthermore, we have discussed the significance of this concept and its effectiveness in downstream tasks including domain generalization and fair learning.
>
> Q3: There is no discussion about the failing points of the proposed method. What happened when the condition variable X and the target Y are entangled in more complex ways? How will the learned representation Z be affected?
>
> A3: We greatly appreciate your suggestion. You make a valid point that when X and Y are highly entangled, the generated Z is prone to failure because H(Y|X) is typically small, making information subtraction more challenging. However, designing experiments to validate this is quite difficult, primarily because there is no universally accepted measurement for the entanglement between variables. The complexity of the relationship between two variables cannot be directly quantified using correlation or mutual information alone. Nevertheless, we agree that in future work, such metrics could be employed to identify the "falling points" you mentioned.
>
> Q4: The formulation and the presented Algorithm are not clear.
>
> A4: Thank you for your suggestions. We have made revisions to Equation (2, 4, 5, 6, 10) and Algorithm (1). If you find any issues with the revised version, we would be grateful if you could specify the detailed concerns. This feedback would enable us to address these points and enhance the clarity and presentation of our manuscript.

---

> ### Author Response · Authors · 2024-11-27
> **Response for the questions**
>
> Here are the responds for the questions.
>
> Q1: Did you try adding a hyper-parameter to one of the terms in the loss? Could that allow for finer control by the user on the learned representation?
>
> A1: Thank you for your valuable feedback. We did incorporate weights into Equation (10) to balance the losses, and we apologize that they were not explicitly stated. It is indeed necessary to revise Equation (10) and conduct a sensitivity analysis. The sensitivity analysis results of Sections 5.1 and 5.3 have been provided in Figure G.1 of Appendix G. The results of other sections will be available in the camera-ready version.
>
> Q2: Did you have issues in training stability?
>
> A2: Yes, as reviewer KedS has noted, MINE is well-known for its instability during training. This is primarily because minor parameter updates in the neural network can cause significant fluctuations in the upper bound estimated by MINE, while simply reducing the learning rate can lead to issues such as slow training. We should acknowledge that careful selection of parameters is required during training, which is almost a universal issue for all frameworks utilizing MINE, and is one of the tradeoffs for enjoying the simplicity of MINE. We have provided the learning rates and network sizes for all the experiments in Table A.1, B.1, C.1, D.1, and E.1.
>
> Q3: Did you test the effectiveness of the proposed method on high dimensional data? How did the computational cost scale?
>
> A3: The focus of our current manuscript is primarily on the theoretical framework. We anticipate exploring very high-dimensional and complex datasets in our future work. In fact, in this work, we have utilized datasets with a dimension greater than 100 for variable X in Section 5.4, and with a dimension of 10 for variable C in Section 5.2.
>
> Regarding computational cost, our framework incorporates two neural networks. The size of the neural network's hidden layers is determined primarily by the complexity of the relationships between the input X, target Y, and conditional variable C, rather than the input dimension itself. Regardless of the size of the input dimension, our architecture remains consistent. Therefore, at this stage, there is no clear relationship between the input dimension and computational cost.
>
> Thank you again for your time and your insightful questions.

---

> > ### Comment · Reviewer_EVYs · 2024-11-27
> >
> > Thanks for the thorough replies.
> > As it stands, I view the manuscript below the level required for publication.
> > I encourage the authors to incorporate the feedback into a revised version and resubmit their work.

---

### Author Response · Authors · 2024-11-27
**Descriptions and comments of the revised version**

Dear reviewers,

First of all, please allow us to express our gratitude for your valuable suggestions. The major feedbacks include:

1. Our contribution is ambiguous: In this manuscript, the major contribution lies in the introduction of Information Subtraction, which is a concept achieved by generating representation for conditional entropy. We have formulated the mathematical expressions necessary to achieve this objective and proposed a neural network architecture to implement it. Finally, we show the effectiveness of our framework in downstream tasks including domain generalization and fair learning.

2. Baseline comparison lacking: The primary focus of the experimental section is to demonstrate that the representation Z we generate performs better in fair learning and domain generalization compared to the original input X, rather than presenting an incremental or SOTA approach.Still, we agree that including baselines for comparison will be beneficial for illustrating the effectiveness of our framework. We are currently in the process of supplementing two contrastive-based baselines, one for discrete scenarios and another for continuous scenarios.

3. Our work v.s. Information estimator: Our work belongs to representation learning, and we use information estimator as one of the modules in the framework. Some information estimators do not emphasize representation for samples in their framework (such as CCMI), and therefore they are different from representation generators.

We have learned a lot from your feedback and revised our manuscripts accordingly. Here are some major modifications. All the modifications have been highlighted in blue.

1. We have made revisions to Equation (2, 4, 5, 6, 10), Figure (3), and Algorithm (1).

2. Detailed experiment settings are provided in Appendix A, B, C, D, and E.

3. We have provided a sensitivity analysis on the hyperparameter in Appendix G.

4. We are still working on the baseline for comparisons. It will be available on the camera-ready version.

Thank you again for your time and your insightful questions.

---

### Meta-Review · Area_Chair_uT2v · 2024-12-18

**Metareview:**

This paper presents a method for learning representations of data which maximize mutual information between inputs and targets while minimizing mutual information between the representation and some given sensitive variables. They present applications in fair machine learning, generalization and other tasks. Unlike prior work this method can be applied to continuous target variables. The method is flexible to many architectures and provided experiments demonstrate the method can work in some settings.

The main issues with the work are its potential novelty, limited experimental validation, potential training instability, clarity, and demonstrated scale.The paper could be improved if the authors address reviewers concerns and added additional (and more challenging) baselines, a deeper discussion on potential stability issues, and an investigation into settings with more complex intra-variable relationships.

Reviewer feedback was consistent and I will recommend rejection.

**Additional Comments On Reviewer Discussion:**

Initially reviewer feedback was consistent recommending rejection for reasons stated in above section. In response the authors responded and committed to; adding baselines, studying sensitivity to hyper-parameters, clarifying math/method, cleaning up the presentation. Throughout rebuttal phase reviewers minds were not changed -- suggesting the changes needed to improve the work would be to great to constitute a minor revision. Thus, reviewer feedback remained consistent, advocating for rejection.

---

### Decision · Program_Chairs · 2025-01-22

Reject